# Factors influencing treatment success in drug-susceptible tuberculosis patients in Ghana: A prospective cohort study

Richard Delali Agbeko Djochie[1,2]◔*, Berko Panyin Anto[1]◔*,
Mercy Naa Aduele Opare-Addo[1]◔, Jonathan Boakye-Yiadom[3]◔

**1** Department of Pharmacy Practice, Faculty of Pharmacy and Pharmaceutical Sciences, College of Health Sciences, Kwame Nkrumah University of Science and Technology, Kumasi, Ghana, **2** Department of Pharmacotherapeutics and Pharmacy Practice, School of Pharmacy and Pharmaceutical Sciences, University of Cape Coast, Cape Coast, Ghana, **3** Department of Epidemiology and Biostatistics, Kwame Nkrumah University of Science and Technology, Kumasi, Ghana

◔ These authors contributed equally to this work.
* richarddjochie@gmail.com (RDAD); bpanto.pharm@knust.edu.gh, berkopanyin@hotmail.com (BPA)

## Abstract

Optimal rates of positive tuberculosis (TB) treatment outcomes are crucial for patient well-being and preventing the emergence of drug-resistant strains. Unfavourable outcomes present significant challenges to healthcare systems worldwide, making it essential to identify factors that influence treatment success. This study aimed to assess treatment outcomes and explore determinants of success to inform effective interventions and contribute to TB eradication efforts. Patients with active TB were monitored from treatment initiation to completion between January 2021 and December 2021. Data were collected using a Redcap-based tool to capture demographics, adverse reactions to antitubercular medications, and treatment outcomes. Quality of life was assessed using the Short-Form 12 version 2 questionnaire at baseline, the second month, and the sixth month. Logistic regression was performed to analyse associations between participant characteristics and treatment success, with odds ratios indicating the strength of associations at a 0.05 significance level. Among 378 participants, 77.3% achieved successful treatment outcomes, while 13.5% were lost to follow-up, 0.5% experienced treatment failure, and 8.7% died. Key factors influencing treatment success included baseline body weight, weight gain during treatment, HIV status, adverse drug reactions, and mental well-being at treatment initiation. Multivariate analysis revealed that gaining at least 3 kg during treatment and having no risk of depression at baseline significantly increased the likelihood of treatment success.

## Introduction

Tuberculosis (TB) remains a global health priority, with over 10 million cases annually and 1.5 million deaths in 2021, making it the second deadliest infectious disease after COVID-19 [1]. The burden is most pronounced in low- and middle-income countries, particularly within the World Health Organisation's (WHO) South-East Asia, Africa, and Western Pacific regions

**Data availability statement:** The data that support the findings of this study are openly available at https://data.mendeley.com/datasets/tdyb9cxxbc/3.

**Funding:** The authors received no specific funding for this work.

**Competing interests:** The authors have declared that no competing interests exist.

[1]. In Ghana, TB prevalence in 2021 stood at 136 cases per 100,000 people, with an estimated 12,000 deaths from TB alone and 3,700 deaths from TB/HIV coinfections [2].

Global treatment success rates for TB vary significantly, with a reported 85% success rate in 2020, falling short of the WHO's End TB Strategy target of 90% [1]. In the African region, this rate averages 79%, and Ghana's rate has plateaued at 84–85% since 2012, despite free treatment availability [3,4]. Regional disparities exist within Ghana, with success rates ranging from 82.5% in the Volta region to over 90% in the Central and Greater Accra regions [5,6]. Optimizing these rates is critical to preventing drug-resistant TB and improving patient outcomes.

Adverse TB treatment outcomes have been linked to factors such as HIV coinfection, treatment nonadherence, alcohol use, drug abuse, stigma, and socioeconomic challenges, including poor housing and malnutrition [7–13]. Additionally, sociodemographic and clinical factors such as low income, unemployment, advanced age, and gender disparities further shape treatment outcomes [12,14,15]. Existing studies in Ghana, often retrospective, provide valuable insights but fail to fully address the role of health-related quality of life and adverse drug reactions (ADRs) in TB treatment success [10,11,16].

Given the persistent gaps in understanding the determinants of TB treatment success, this study sought to explore the factors influencing outcomes among drug-susceptible TB patients in Ghana, with a focus on ADRs and health-related quality of life. By providing a comprehensive analysis, the study aimed to inform tailored interventions for improving treatment success rates, thereby reducing TB's burden in Ghana and similar global settings.

## Materials and methods

### Study design and participants

This was a prospective cohort study conducted at eight TB treatment centres in the Eastern and Ashanti Regions of Ghana. The study sites were selected based on their consistent reporting of the highest annual TB case volumes over the past five years, ensuring a representative and diverse patient population. The facilities included three primary care hospitals and one regional referral hospital, capturing various levels of healthcare delivery. Participants were followed from the initiation of treatment to its completion at six months.

The study population consisted of newly diagnosed patients with drug-susceptible tuberculosis (TB) who were prescribed the standard treatment regimen of rifampicin, ethambutol, isoniazid, and pyrazinamide. The drug-susceptible status of their TB was confirmed using the routine GeneXpert test, which is standard practice for diagnosing TB in Ghana. Participants were required to meet the following inclusion criteria: being 18 years of age or older, having access to a mobile phone for follow-up communication, and providing informed consent. Patients with known psychiatric conditions or significant liver or kidney impairments were excluded to reduce confounding factors that could independently influence treatment outcomes.

To evaluate health-related quality of life (HRQOL), participants completed the Short-Form 12 version 2 (SF-12v2) survey questionnaire at baseline (prior to treatment initiation), at month two, and at the end of treatment (month six) [17–19]. Adverse drug reactions to anti-TB medications were monitored through weekly phone calls and by reviewing ADR reports recorded in treatment cards during clinic visits.

### Data collection and analysis

The data collection instrument for this study was developed using REDCap, aligned with study objectives, validated, and pretested at a non-study hospital. Feedback from the pre-test

informed necessary adjustments. Participant recruitment and data collection occurred from January 10, 2021, to December 15, 2021. Trained research assistants contacted participants weekly to monitor adverse effects from TB treatment. Adverse drug reactions (ADRs) recorded in treatment cards were extracted, and a literature-based side effect checklist was used for screening. The Naranjo ADR Probability Scale determined causality, with ADRs classified as Certain (≥9), Probable (5–8), or Possible (1–4) included in the analysis [20].

ADR severity was assessed using Hartwig's severity scale [21,22]:

- Severe (levels 5–7): Death, hospitalization, or prolonged hospital stay.

- Moderate (levels 3–4): Required treatment withholding, discontinuation, or antidotes.

- Mild (levels 1–2): Required suspending treatment without antidotes.

For analysis, moderate-to-severe ADRs were grouped as major ADRs, and mild ADRs as minor ADRs.

The SF12v2 tool evaluated physical (PCS) and mental (MCS) components, with scores ranging from 0–100, where higher scores indicate better quality of life. A minimum clinically important difference (MCID) was defined as a 3-point increase in PCS or MCS at treatment completion. A physical component summary (PCS) score below 47.9 (mean population norm) indicated physical health impairment while an MCS score below 42 indicated risk of depression [18].

## Statistical analysis

Data cleansing was performed using Microsoft Excel 2016, followed by data analysis with STATA version 17. Results are presented as frequencies, percentages, and means with standard deviations. Chi-squared or Fisher's exact tests were used to identify socio-demographic and clinical factors associated with treatment success. Logistic regression, with crude and adjusted odds ratios, assessed the strength of associations between patient characteristics and treatment success. Statistical significance was set at $p < 0.05$.

The data that support the findings of this study are openly available in Mendeley Data at https://data.mendeley.com/datasets/tdyb9cxxbc/3. For this analysis, only participants with complete data were included, while those with missing data were excluded.

## Ethical considerations

The study was conducted in accordance with ethical guidelines and received approval from the Ghana Health Service Ethics Review Committee (approval number GHS-ERC 007/11/20) on January 2, 2021.

Ethical principles, including voluntary participation, beneficence, nonmaleficence, privacy, and confidentiality, were upheld throughout the study. Written informed consent was obtained from participants, who retained the right to withdraw at any time without explanation.

## Results

### Sociodemographic characteristics of participants

Of the 378 participants, most were male (67.2%), married (44.2%), and employed (75%), with an average age of 45.3 years (±15.1). As detailed in Table 1, 12.9% had no formal education. A minority had extrapulmonary tuberculosis (6.1%), identified as smokers (6.4%), reported alcohol use (25.4%), or tested positive for HIV (25.7%).

**Table 1. Characteristics of Study Participants.**

| Attributes | Number (n) | Proportion (%) |
|---|---|---|
| Age groups (years) | | |
| 18–25 | 26 | 6.9 |
| 26–35 | 84 | 22.2 |
| 36–45 | 90 | 23.8 |
| 46–55 | 90 | 23.8 |
| 56–65 | 49 | 13.0 |
| ≥66 | 39 | 10.3 |
| Sex | | |
| Male | 254 | 67.2 |
| Female | 124 | 32.8 |
| Marital status | | |
| Married | 167 | 44.2 |
| Single | 127 | 33.6 |
| Divorced | 41 | 10.9 |
| Widowed | 43 | 11.3 |
| Employment status | | |
| Employed | 289 | 76.5 |
| Unemployed | 89 | 23.5 |
| Educational level | | |
| Basic | 230 | 60.9 |
| Secondary | 70 | 18.5 |
| Tertiary | 29 | 7.7 |
| No formal education | 49 | 12.9 |
| Tuberculosis Type | | |
| Pulmonary TB | 350 | 92.6 |
| Extra-pulmonary TB | 23 | 6.1 |
| Both Types | 5 | 1.3 |
| Smoking Behaviour | | |
| Smoker | 24 | 6.4 |
| Non-smoker | 354 | 93.6 |
| Alcohol Consumption | | |
| Yes | 96 | 25.4 |
| No | 282 | 74.6 |
| HIV Status | | |
| HIV positive | 97 | 25.7 |
| HIV negative | 275 | 72.8 |
| Unknown status | 6 | 1.5 |
| Baseline HRQOL | | |
| Risk of Depression | 97 | 25.7 |
| Physical impairment | 298 | 78.8 |
| Adverse Drug Reaction | | |
| None | 198 | 52.4 |
| Minor | 68 | 18.0 |
| Major | 112 | 29.6 |

HIV = Human Immunodeficiency Virus; HRQOL = Health-related Quality of Life; TB = Tuberculosis

Among 378 patients observed over 51,730 person-months, 181 (47.9%) reported at least one adverse drug reaction (ADR), totaling 904 events and yielding an overall incidence rate of 1.75 per 100 person-months. The nervous system was the most affected, with 155 events, followed by the genitourinary system (124 events) and the gastrointestinal system (123 events). All participants initially completed the baseline SF-12v2 questionnaire, but by the end of the intensive phase, 335 participants (88.6%) had completed it. This number further declined to 294 (77.8%) by the end of treatment, primarily due to treatment default, withdrawal from the study, or death.

## Determinants of treatment success among study participants

The overall tuberculosis treatment success rate among participants was 77.3%, which included those who were cured (44.2%) and those who completed therapy (33.1%). Adverse outcomes included 13.5% lost to follow-up, 8.7% deaths, and 0.5% classified as treatment failures.

Several factors significantly influenced treatment success, as detailed in Table 2. Notably, participants with a baseline weight of at least 53 kg or those who gained at least 3 kg during treatment were significantly more likely to achieve success compared to their counterparts. Baseline physical and mental health were also important predictors, with better health status correlating with higher success rates. For instance, participants with good physical health achieved an 88.8% success rate, compared to 74.2% for those with poor physical health.

HIV status was a significant determinant of treatment outcomes. HIV-negative participants had a higher treatment success rate (81.8%) compared to HIV-positive participants (64.9%, $p < 0.001$). Among HIV-positive individuals, those with HIV-1 infection had better success rates than those with HIV-2 infection ($p = 0.019$). Additionally, participants with extrapulmonary TB achieved a 100% success rate, compared to 75.4% for those with pulmonary TB.

Baseline health-related quality of life (HRQOL) also influenced treatment outcomes. Participants with good baseline physical health were more likely to achieve treatment success than those with impaired physical health ($p = 0.014$). Similarly, good baseline mental health was associated with higher odds of treatment success ($p = 0.027$).

Adverse drug reactions (ADRs) affected treatment success in specific ways. While the overall presence or absence of ADRs and its severity did not impact outcomes, participants who did not experience gastrointestinal ADRs had significantly higher odds of success ($p = 0.019$). Those without hepatobiliary ADRs had even greater odds of achieving treatment success ($p = 0.005$).

Several patient characteristics emerged as significant determinants of treatment success in the bivariate logistic regression analysis (Table 3). Baseline weight and weight gain during treatment were strongly associated with outcomes. Participants with a baseline weight above 53 kg and those who gained at least 3 kg during treatment demonstrated higher odds of treatment success.

A multivariate logistic regression analysis was conducted to identify independent predictors of tuberculosis treatment success, accounting for all significant factors identified in earlier analyses. The results are presented in Table 4.

Two factors emerged as significant independent predictors of treatment success: gaining at least 3 kg during treatment (Adjusted Odds Ratio [AOR]: 62.7, 95% CI: 3.77–1,041.97, $p = 0.004$) and not being at risk of depression at baseline (AOR: 0.12, 95% CI: 0.02–0.67, $p = 0.016$).

Other factors, such as gastrointestinal adverse drug reactions (ADRs), type of HIV infection, and baseline physical health, did not show significant associations in the multivariate analysis.

**Table 2.  Association of Patient Characteristics with Treatment Success.**

| Parameter | N | % with Treatment Success | p-value |
|---|---|---|---|
| Gender | | | |
| Male | 254 | 78.7 | 0.322 |
| Female | 124 | 74.2 | |
| Weight at Baseline | | | |
| < 53 kg | 219 | 72.6 | **0.011** |
| ≥ 53 kg | 159 | 83.6 | |
| Total Weight Gained | | | |
| < 3 kg | 174 | 65.5 | **<0.001** |
| ≥ 3 kg | 170 | 90.9 | |
| Alcohol Use | | | |
| Yes | 96 | 72.9 | 0.241 |
| No | 282 | 78.7 | |
| Tobacco Smoking | | | |
| Yes | 24 | 83.3 | 0.462 |
| No | 354 | 76.8 | |
| HIV Status | | | |
| HIV-positive | 97 | 64.9 | **0.002** |
| HIV-negative | 275 | 81.8 | |
| HIV status unknown | 6 | 66.7 | |
| HIV Type | | | |
| HIV-1 | 83 | 69.9 | **0.011** |
| HIV-2 | 7 | 57.1 | |
| Both HIV-1 & 2 | 7 | 14.3 | |
| Site of TB Infection | | | |
| Pulmonary | 350 | 75.4 | **0.003** |
| Extra-pulmonary | 28 | 100.0 | |
| HAART Before TB Treatment | | | |
| Yes | 84 | 72.0 | 0.133 |
| No | 13 | 57.4 | |
| ADR status | | | |
| No ADR | 198 | 77.3 | 0.991 |
| At least one ADR | 180 | 77.2 | |
| Baseline Physical HRQOL | | | |
| Good Physical Health | 76 | 88.8 | **0.006** |
| Poor Physical Health | 302 | 74.2 | |
| Baseline Mental HRQOL | | | |
| Good Mental Health | 281 | 80.2 | **0.026** |
| Poor Mental Health | 97 | 69.1 | |

1. **Acronyms:** ADR = adverse drug reaction; HAART = highly active antiretroviral treatment; HIV = human immunodeficiency virus; HRQOL = Health-related Quality of Life; TB = tuberculosis; WHO = World Health Organization.

2. **Statistical Highlight:** Boldface p-value entries indicate statistically significant associations.

## Discussion

The tuberculosis (TB) treatment success rate in this study was 77.3%, comprising 44.2% of participants who were cured and 33.1% who completed their treatment. This rate is below the national average of 84% [2,5,23] and the WHO target of 90% [24]. It also trails the success

Table 3. Impact of Participants' Characteristics on the Odds of Treatment Success.

| Participant characteristics | Odds of Treatment Success | | |
|---|---|---|---|
| | OR | 95% CI | P-value |
| Weight at Baseline | | | |
| ≤ 53 kg | 1 | 1.2–3.2 | **0.012** |
| >53 kg | 1.9 | | |
| Total Weight Gained | | | |
| <3 kg | 1 | 2.2–9.0 | **<0.001** |
| ≥3 kg | 4.5 | | |
| HIV status | | | |
| HIV-positive | 1 | 1.4–4.1 | **<0.001** |
| HIV-negative | 2.4 | | |
| Type of HIV | | | |
| HIV-2 | 1 | 1.3–3.7 | **0.019** |
| HIV-1 | 4.2 | | |
| Baseline Physical HRQOL | | | |
| Poor physical health | 1 | 1.2–5.3 | **0.014** |
| Good physical health | 2.5 | | |
| Baseline Mental HRQOL | | | |
| Poor mental health | 1 | 1.1–3.0 | **0.027** |
| Good mental health | 1.8 | | |
| Gastrointestinal ADR | | | |
| Yes | 1 | 1.1–3.0 | **0.019** |
| No | 1.8 | | |
| Hepatobiliary System ADR | | | |
| Yes | 1 | 2.6–183.0 | **0.005** |
| No | 21.8 | | |

1. **Acronyms:** ADR = Adverse Drug Reaction; HIV = Human Immunodeficiency Virus; HRQOL = Health-related Quality of Life; OR = Odds Ratio

2. **Statistical Highlight:** Boldface p-values indicate statistically significant associations.

3. **Interpretation Guidance:** Odd ratios greater than 1 indicate higher likelihood, while those less than 1 indicate lower likelihood of achieving tuberculosis treatment success compared to the reference group.

Table 4. Logistic Regression Analysis of Predictors of Tuberculosis Treatment Success.

| Predictor | Standard Error | z | AOR | 95% CI | p-value |
|---|---|---|---|---|---|
| Weight Gain ≥3 kg | 89.91 | 2.89 | 62.7 | 3.77–1,041.97 | **0.004** |
| Risk of Depression | 0.10 | −2.41 | 0.12 | 0.02–0.67 | **0.016** |
| Gastrointestinal ADRs (Yes) | 0.51 | −0.67 | 0.52 | 0.08–3.51 | 0.502 |
| HIV Type 2 | 0.14 | −−1.61 | 0.09 | 0.005–1.68 | 1.07 |
| Baseline Physical Weakness | 3.31 | 1.00 | 3.00 | 0.35–26.03 | 0.318 |
| Constant | 3.99 | 1.17 | 3.62 | 0.42–31.42 | 0.244 |

1. Acronyms: ADR = Adverse Drug Reaction; AOR = Adjusted Odds Ratio; HIV = Human Immunodeficiency Virus; CI = Confidence Interval

2. **Statistical Highlight:** Boldface p-values indicate statistically significant associations.

3. **Interpretation Guidance:** Adjusted Odd Ratios greater than 1 indicate higher likelihood, while those less than 1 indicate lower likelihood of achieving treatment success in the presence of other predictors.

rates of 81.8%, and 81.0% reported in studies conducted in Somalia [25] and India [26] respectively. However, it exceeds the rates of 71.4% and 60.1% reported in Ethiopian studies [27,28] and the 68.5% from a 10-year retrospective study in Ghana's Ashanti Region [16].

Adverse treatment outcomes occurred in 22.8% of participants, including 13.5% who were lost to follow-up, 8.7% who died, and 0.5% who experienced treatment failure. The loss to follow-up rate aligns with the 6.2% reported in Somalia [25] and the 8% in Ethiopia [27] but is lower than the 21.3% [28] and 12.5% [26] reported in other studies. Nonetheless, it exceeds the 0.9% reported by Otchere et al. (2018) in Ghana. Addressing treatment default requires robust pre-treatment adherence counselling and consistent reinforcement during clinic visits, as emphasized by Muture et al. (2011). This study did not find significant associations between TB treatment default rates and factors such as sex (p = 0.874), marital status (p = 0.190), educational level (p = 0.119), baseline physical health impairment (p = 0.177), baseline mental health impairment (p = 0.049), or age (p = 0.908). While previous studies [8,27] have reported that age, gender, and employment status influence TB treatment default rates, these associations were not observed in this study. This discrepancy may be attributed to the relatively low number of defaulters in the sampled population. Further research with larger sample sizes is needed to validate these findings and investigate these potential relationships more comprehensively. Among participants who died (n = 33), 63.6% were HIV-positive, underscoring the vulnerability of this subgroup. The death rate (8.7%) was below the TB strategic plan's 10% target [29] and rates of 17.7% and 10.1% reported in Ethiopia [28,30]. However, it exceeded the rate in India (2%; Jain et al., 2012). HIV accelerates TB progression and increases susceptibility to opportunistic infections [31]. Poor treatment adherence due to pill burden and drug interactions further reduces success rates in HIV-positive individuals [8,25,28,32]. This highlights the importance of early HIV diagnosis, integrated TB-HIV care, and psychological support during treatment.

This study found that HIV type influences outcomes, with HIV-1 patients having more than double the odds of treatment success compared to HIV-2 patients. This aligns with a prospective study from Guinea Bissau [31], underscoring the need for tailored approaches to managing TB in HIV patients based on subtype.

Interestingly, all participants with extrapulmonary TB achieved treatment success, consistent with findings from Bhutan [33] and Ethiopia [34]. This diverges from studies in other African regions [27,28], suggesting a complex relationship that warrants further investigation.

Weight gain emerged as a significant predictor of treatment success. Patients who gained at least 3 kg during treatment or had a baseline weight above 53 kg were more likely to succeed, mirroring findings from Peru [35] and Ethiopia [36]. Improved nutrition strengthens the immune system and enhances recovery, emphasizing the need for nutritional interventions for vulnerable populations, such as those facing homelessness or poverty [37–39].

Mental health at baseline emerged as a significant determinant of treatment success. Participants without depression at the start of treatment had nearly twice the odds of achieving favourable outcomes. This aligns with findings from studies that highlight the critical role of mental health in TB treatment outcomes [40,41]. Integrating psychosocial counselling into TB treatment plans is essential, as psychological distress often correlates with poorer recovery rates [42]. Furthermore, addressing stigma within communities can create a supportive environment that enhances patient adherence and recovery [43].

Similarly, physical impairment at baseline was associated with poorer treatment outcomes. This underscores the importance of managing comorbidities and implementing targeted interventions to improve patients' physical health, which can significantly enhance their treatment success.

The absence of ADRs was associated with better outcomes, consistent with studies in India [44] and China [45]. Participants without gastrointestinal ADRs were twice as likely to

achieve treatment success, and those without hepatobiliary ADRs had 22 times higher odds of success. While ADR severity did not significantly influence outcomes in this study, these findings suggest the need for vigilant monitoring and management of ADRs to enhance treatment efficacy.

This study underscores weight gain and mental health as strong predictors of TB treatment success, advocating for comprehensive care that integrates nutritional, psychological, and medical support. While ADRs and HIV subtype require further investigation, these findings highlight the value of multidisciplinary approaches in optimizing TB treatment outcomes globally.

The study observed that successful treatment outcomes were more common among males, tobacco non-smokers, alcohol non-drinkers, and patients on HAART, though these differences were not statistically significant ($p > 0.05$). Additionally, employment status ($p = 0.169$), marital status ($p = 0.772$), sex ($p = 0.322$), level of education ($p = 0.313$) and age ($p = 0.376$) showed no significant association with treatment outcomes. These findings suggest that while these factors may contribute to treatment success, they do not independently determine outcomes in this study population. However, previous studies [13,30,46–48] have reported significant associations between gender, substance use, and HAART with treatment outcomes. This discrepancy underscores the need for further research to clarify these relationships and explore factors that may account for variations across different study populations.

## Study limitations

This study faced some limitations, including potential recall bias from self-reported data and a 13.5% loss to follow-up. The six-month follow-up may not capture long-term outcomes, and small subgroup sizes, such as participants with extrapulmonary TB, limited statistical power. Nonetheless, the study offers valuable insights into key determinants of tuberculosis treatment success, providing evidence to guide interventions and improve patient care.

## Conclusion

This study highlights key predictors of tuberculosis treatment success, including weight gain during treatment and baseline mental health status. Participants who gained at least 3 kg and those without depression at the start of treatment had significantly higher odds of success. The findings underscore the importance of integrating nutritional support and psychosocial counselling into TB care. Although adverse drug reactions and HIV subtype were not statistically significant, further research is needed to explore their potential impact. Overall, a comprehensive approach addressing both physical and mental health is essential for improving TB treatment outcomes.

## Author contributions

**Conceptualization:** Richard Delali Agbeko Djochie, Berko Panyin Anto, Mercy Naa Aduele Opare-Addo.

**Data curation:** Richard Delali Agbeko Djochie, Jonathan Boakye-Yiadom.

**Formal analysis:** Richard Delali Agbeko Djochie, Berko Panyin Anto, Mercy Naa Aduele Opare-Addo, Jonathan Boakye-Yiadom.

**Investigation:** Richard Delali Agbeko Djochie.

**Methodology:** Richard Delali Agbeko Djochie, Berko Panyin Anto, Mercy Naa Aduele Opare-Addo.

**Supervision:** Berko Panyin Anto.

**Visualization:** Jonathan Boakye-Yiadom.

**Writing – original draft:** Richard Delali Agbeko Djochie, Berko Panyin Anto, Mercy Naa Aduele Opare-Addo, Jonathan Boakye-Yiadom.

**Writing – review & editing:** Richard Delali Agbeko Djochie, Berko Panyin Anto, Mercy Naa Aduele Opare-Addo, Jonathan Boakye-Yiadom.

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
