## [Decision Letter · Decision Letter 0]

30 Dec 2024

PGPH-D-24-02857

Factors Influencing Treatment Success in Drug-Susceptible Tuberculosis Patients in Ghana: A Prospective Cohort Study

Dear Dr. Djochie,

Thank you for submitting your manuscript to PLOS Global Public Health. After careful consideration, we feel that it has merit but does not fully meet PLOS Global Public Health’s publication criteria as it currently stands. Therefore, we invite you to submit a revised version of the manuscript that addresses the points raised during the review process.

We look forward to receiving your revised manuscript.

Kind regards,

Lei Gao

Academic Editor

Journal Requirements:

Additional Editor Comments (if provided):

1. The drug resistance status of tuberculosis patients should be analyzed or discussed。

2. In table 4, I would like to suggest using HIV status but not HIV-2 infection as potential predictor.

Reviewers' comments:

Reviewer's Responses to Questions

**Comments to the Author**

1. Does this manuscript meet PLOS Global Public Health’s publication criteria ? Is the manuscript technically sound, and do the data support the conclusions? The manuscript must describe methodologically and ethically rigorous research with conclusions that are appropriately drawn based on the data presented.

Reviewer #1: Yes

Reviewer #2: Partly

2. Has the statistical analysis been performed appropriately and rigorously?

Reviewer #1: I don't know

Reviewer #2: Yes

3. Have the authors made all data underlying the findings in their manuscript fully available (please refer to the Data Availability Statement at the start of the manuscript PDF file)?

Reviewer #1: Yes

Reviewer #2: Yes

4. Is the manuscript presented in an intelligible fashion and written in standard English?

Reviewer #1: Yes

Reviewer #2: Yes

5. Review Comments to the Author

Reviewer #1: Important analysis and findings that will be applicable in the field and with policy makers. Can you say more about the need to provide additional services to people living with HIV given the less positive outcomes in this group?

Reviewer #2: The following comment are better outlined in the table available in attachment.

93 -To evaluate HRQOL, participants completed the Short-Form 12 version 2 (SF-12v2) survey

Comment: Please write HRQOL in full for the first in-text mention, abbreviate in brackets and then continue with the abbreviation.

196-198 Conversely, gender, alcohol use, tobacco smoking, and adverse drug reactions (ADRs) did not significantly affect treatment outcomes. Notably, treatment success rates for participants with and without ADRs were nearly identical, and the severity of ADRs did not influence outcomes.

Comment: This belongs to the Discussion section, comparing the results with those of other studies conducted in similar settings with almost similar socioeconomic statuses.

Line 211-218 -HIV status also played a critical role. HIV-negative participants had significantly greater odds of achieving treatment success compared to their HIV-positive counterparts. Among HIV-positive individuals, those with HIV-1 infection were substantially more likely to succeed compared to those with HIV-2 infection. Health-related quality of life at the start of treatment further influenced outcomes. Good baseline physical and mental health were both associated with increased odds of treatment success. Adverse drug reactions impacted treatment success in specific ways. Participants who did not experience gastrointestinal ADRs had higher odds of success, while those without hepatobiliary ADRs had markedly greater odds of achieving treatment success.

Comment: This section belongs to the Discussion as well, where findings are compared with those of other studies. The Results section should strictly present results without including the author's perceptions of the findings, as these are reserved for the Discussion section. While the author can elaborate on the results in this section, it should be a concise paragraph that directly reflects the data presented in the table, avoiding any interpretative commentary.

Line: 139- Of the 378 participants, most were male (67.2%), married (44.2%), and employed (75%), with an...

Comment: While reading the Results section, I anticipated a discussion on the significance of marital status in relation to factors contributing to treatment success. However, this was not addressed in the Discussion section, even though stigma and mental health status were mentioned. Since marital status was included in the report form in REDCap and analyzed, it would be valuable to include a discussion on this variable. Different studies have explored the relationship between family structure, support, marital status, and treatment compliance. This could also be compared with treatment success versus lost to follow-up, discussing whether the authors' findings align with or contradict findings from other studies.

Line: 158- Adverse outcomes included 13.5% lost to follow-up…

Comment: The 13.5% lost to follow-up is highly significant and warrants discussion as it directly impacts poor anti-TB therapy outcomes. It would be beneficial to explore their marital status, employment status, and baseline HRQOL. Rather than focusing on individual factors, the discussion should emphasize commonalities among these individuals, providing insights into shared characteristics that might contribute to this outcome. This is just a suggestion.

Overall, the manuscript is well written, with only minor adjustments needed. However, the study has collected rich data, and it may be worthwhile to discuss a broader range of the data collected.

6. PLOS authors have the option to publish the peer review history of their article (what does this mean? ). If published, this will include your full peer review and any attached files.

**Do you want your identity to be public for this peer review?** For information about this choice, including consent withdrawal, please see our Privacy Policy .

Reviewer #1: No

Reviewer #2: **Yes: ** Sthabiso Bohlela

---

## [Editor Report · Decision Letter 1]

27 Jan 2025

Factors Influencing Treatment Success in Drug-Susceptible Tuberculosis Patients in Ghana: A Prospective Cohort Study

PGPH-D-24-02857R1

Dear Dr Djochie,

We are pleased to inform you that your manuscript 'Factors Influencing Treatment Success in Drug-Susceptible Tuberculosis Patients in Ghana: A Prospective Cohort Study' has been provisionally accepted for publication in PLOS Global Public Health.

Best regards,

Lei Gao

Academic Editor